# Trabid patient mutations impede the axonal trafficking of adenomatous polyposis coli to disrupt neurite growth

Daniel Frank[1,2†], Maria Bergamasco[2,3†], Michael J Mlodzianoski[2,4†], Andrew Kueh[2,5,6,7], Ellen Tsui[2,8], Cathrine Hall[2,9], Georgios Kastrappis[10], Anne Kathrin Voss[2,3], Catriona McLean[11], Maree Faux[12], Kelly L Rogers[2,4], Bang Tran[10], Elizabeth Vincan[10,13], David Komander[1,2], Grant Dewson[1,2]*, Hoanh Tran[1,2,10]*

[1]Ubiquitin Signalling Division, Walter and Eliza Hall Institute of Medical Research, Parkville, Australia; [2]Department of Medical Biology, The University of Melbourne, Parkville, Australia; [3]Epigenetics and Development Division, Walter and Eliza Hall Institute of Medical Research, Parkville, Australia; [4]Centre for Dynamic Imaging, Walter and Eliza Hall Institute of Medical Research, Parkville, Australia; [5]Melbourne Advanced Genome Editing Centre, Walter and Eliza Hall Institute of Medical Research, Parkville, Australia; [6]Olivia Newton-John Cancer Research Institute, Heidelberg, Australia; [7]School of Cancer Medicine, La Trobe University, Heidelberg, Australia; [8]Histology Facility, Walter and Eliza Hall Institute of Medical Research, Parkville, Australia; [9]Inflammation Division, Walter and Eliza Hall Institute of Medical Research, Parkville, Australia; [10]Department of Infectious Diseases, The University of Melbourne at The Peter Doherty Institute for Infection and Immunity, Melbourne, Australia; [11]Department of Anatomical Pathology, The Alfred Hospital, Melbourne, Australia; [12]Neuro-Oncology Group, Murdoch Children's Research Institute, Parkville, Australia; [13]The Victorian Infectious Diseases Reference Laboratory, Royal Melbourne Hospital at The Peter Doherty Institute for Infection and Immunity, Melbourne, Australia

*For correspondence:
dewson@wehi.edu.au (GD);
hoanh.tran@unimelb.edu.au (HT)

†These authors contributed equally to this work

**Abstract** *ZRANB1* (human Trabid) missense mutations have been identified in children diagnosed with a range of congenital disorders including reduced brain size, but how Trabid regulates neurodevelopment is not understood. We have characterized these patient mutations in cells and mice to identify a key role for Trabid in the regulation of neurite growth. One of the patient mutations flanked the catalytic cysteine of Trabid and its deubiquitylating (DUB) activity was abrogated. The second variant retained DUB activity, but failed to bind STRIPAK, a large multiprotein assembly implicated in cytoskeleton organization and neural development. *Zranb1* knock-in mice harboring either of these patient mutations exhibited reduced neuronal and glial cell densities in the brain and a motor deficit consistent with fewer dopaminergic neurons and projections. Mechanistically, both DUB-impaired and STRIPAK-binding-deficient Trabid variants impeded the trafficking of adenomatous polyposis coli (APC) to microtubule plus-ends. Consequently, the formation of neuronal growth cones and the trajectory of neurite outgrowth from mutant midbrain progenitors were severely compromised. We propose that STRIPAK recruits Trabid to deubiquitylate APC, and that in cells with mutant Trabid, APC becomes hyperubiquitylated and mislocalized causing impaired organization of the cytoskeleton that underlie the neuronal and developmental phenotypes.

## eLife assessment

This study defines the roles for two different missense mutations observed in patients in the Trabid/ZRANB1 gene associated in children with a range of congenital disorders including reduced brain size. The study is **important** because the findings have theoretical or practical implications beyond a single subfield, as the study of DUB and cytoskeletal alterations have implications for neurodevelopment broadly. The methods are **convincing** as they utilize appropriate and validated methodology in line with current state-of-the-art by incorporating knock-in mice of the patient mutations. Many of the reviewer comments were focused on potential next experiments, rather than on evaluation of the data at hand, and the authors have considered these as future studies. The work as presented suggests critical roles for Trabid in the STRIPAK complex mediating APC deubiquitylation.

## Introduction

Deubiquitylating enzymes (DUBs) are integral components of the ubiquitin system that control protein functions essential for healthy development and aging (*Clague et al., 2019*). DUBs catalyze the removal of ubiquitin from protein substrates to regulate protein stability, activity, interaction, or localization (*Komander et al., 2009*). Ubiquitin can be assembled into polymers linked through one of eight internal ubiquitin residues – Met1, Lys6, Lys11, Lys27, Lys29, Lys33, Lys48, and Lys63 – and impairment of DUB function can lead to the accumulation of one or more of these ubiquitin chain linkages on a protein substrate with deleterious consequences. For example, LINKage-specific deubiquitylation deficiency-induced Embryonic Defects (LINKED), a human syndrome caused by loss-of-function mutations in the DUB OTUD5, is associated with increased Lys48-linked ubiquitylation and turnover of chromatin remodeling and transcription factors that are critical for normal embryonic development (*Beck et al., 2021*). Consequently, LINKED syndrome patients manifest severe congenital malformations and die in early infancy (*Beck et al., 2021*). OTUD5 belongs to a 17-member family of human DUBs that possess an OTU (ovarian tumor) catalytic domain and include the closely related OTU DUBs A20, Cezanne, and Trabid. Trabid missense mutations have been identified in children diagnosed with a range of developmental disorders including microcephaly (*Deciphering Developmental Disorders, 2015*).

Trabid has two defining biochemical features. First, it exhibits strong DUB specificity for Lys29- and Lys33-linked ubiquitin chains, cleaving these chain types 40-fold more efficiently than Lys63-linked chains (*Licchesi et al., 2011*). Lys29-linked polyubiquitin exists mainly as heterotypic chains containing Lys48 linkages that target substrates for degradation (*Kristariyanto et al., 2015*; *Leto et al., 2019*; *Harris et al., 2021*). Trabid is thought to modulate the levels of Lys29/Lys48 mixed chains to regulate proteostasis, autophagy, and cell division (*Chen et al., 2021*; *Yu et al., 2021*; *Vaughan et al., 2022*). Lys33-linked ubiquitin polymers have been implicated in post-Golgi protein trafficking (*Yuan et al., 2014*), and Lys63-linked chains are abundant cellular adducts with established roles in the sorting of endosomal/lysosomal cargo (*Erpapazoglou et al., 2014*) and mediating the formation of protein assemblies (*Tran and Polakis, 2012*). The second defining feature of Trabid is that it binds the Striatin-interacting phosphatase and kinase (STRIPAK) complex (*Sowa et al., 2009*; *Tran et al., 2013*; *Harris et al., 2021*), a large multiprotein assembly implicated in cytoskeleton organization, cell migration, and neural development (*Hwang and Pallas, 2014*; *Sakuma et al., 2014*; *Madsen et al., 2015*; *Bazzi et al., 2017*; *Kück et al., 2019*).

We have previously identified the APC protein as a candidate Trabid substrate (*Tran et al., 2008*). Trabid knockdown in HEK293 cells caused APC to become modified with polyubiquitin and aggregate in the cytoplasm, whereas high Trabid levels correlated with hypoubiquitylated APC that accumulated in the membrane protrusions of long cell processes. These observations led us to propose that Trabid regulates the ubiquitylation and subcellular localization of APC (*Tran et al., 2013*). In neurons, APC organizes the cortical cytoskeleton to promote the formation of growth cones and the steering of growing axons that are essential for embryonic brain development (*Yokota et al., 2009*; *Preitner et al., 2014*; *Dogterom and Koenderink, 2019*; *Efimova et al., 2020*). High expression of Trabid, APC, and STRIPAK proteins in neural stem cells (*Castets et al., 2000*; *Blanpain et al., 2004*; *Yokota et al., 2009*) indicate that Trabid's association with STRIPAK and its ability to regulate APC polyubiquitylation may be important for neurogenesis. We now show that two Trabid mutants found in children with developmental deficits are impaired in two distinct biochemical activities that culminate in the

perturbed trafficking of APC to neurite tips. We propose that the neuronal and developmental pheno-types associated with these Trabid loss-of-function mutations are primarily caused by the mislocaliza-tion of APC that leads to defective cytoskeleton organization and aberrant cell locomotion.

## Results

### Trabid patient variants are impaired in DUB activity and STRIPAK binding

*Trabid/ZRANB1* patient missense mutations, R438W and A451V, are linked to developmental micro-cephaly (***Deciphering Developmental Disorders, 2015***). Mapping these residues onto the domain structure of Trabid shows that they flank either side of Trabid's catalytic cysteine C443 (***Figure 1A***). Residue R438 projects prominently into the catalytic cleft formed by the ankyrin repeats and the OTU core of Trabid's catalytic domain (***Licchesi et al., 2011***), whereas residue A451 resides at the back of the active site, opposite to C443 on alpha helix 4 (***Figure 1B***). To determine if patient mutations R438W and A451V influence substrate catalysis, we purified the mutant AnkOTU domains and tested their ability to hydrolyze synthetic ubiquitin chains in vitro (***Licchesi et al., 2011***). Wild-type Trabid AnkOTU generated appreciable amounts of mono-ubiquitin within 15 min incubation with either Lys29- or Lys63-linked di-ubiquitin chains (***Figure 1C and D***). In contrast, Trabid R438W showed near total loss of DUB activity, and cleaved ubiquitin product was only readily detected after a prolonged 120 min incubation (***Figure 1C and D***). A catalytically inactive C443S mutant AnkOTU domain failed to hydrolyze ubiquitin chains even after 2 h incubation (***Figure 1C and D***). Interestingly, the Trabid A451V AnkOTU domain cleaved Lys29- or Lys63-linked di-ubiquitin with comparable efficiency to wild-type Trabid, despite it being slightly less stable as a recombinant protein based on thermal stability assay (***Figure 1—figure supplement 1A***). A ubiquitin suicide probe assay revealed that R438W and A451V Trabid AnkOTU proteins retained a functional catalytic interaction with ubiquitin (***Figure 1—figure supplement 1B***).

In cells, the inability of the DUB-inactive Trabid C443S to cleave polyubiquitin led to stable inter-action of this mutant with polyubiquitylated substrates (***Tran et al., 2008***; ***Licchesi et al., 2011***). We, therefore, asked whether the DUB-impaired R438W Trabid likewise exhibited increased binding to polyubiquitin in cells. Polyubiquitin chains readily co-precipitated with both FLAG-tagged Trabid R438W and C443S expressed in HEK293T cells, consistent with the compromised DUB activity of these mutants (***Figure 1E***). The FLAG-Trabid A451V variant did not co-precipitate abundant polyubiq-uitylated substrates, consistent with it retaining full DUB activity (***Figure 1E***).

We have previously identified an interaction between Trabid and STRIPAK that is important for the deubiquitylation of substrate APC (***Tran et al., 2013***). Whilst A451V retained full DUB activity, strik-ingly, in contrast to FLAG-Trabid wild-type, R438W, and C443S, the FLAG-Trabid A451V mutant failed to efficiently co-precipitate several STRIPAK components, including Striatin3 and STRIP1 (***Figure 1E***). All FLAG-Trabid proteins co-precipitated similar levels of the substrate E3 ubiquitin ligase HECTD1 (***Tran et al., 2013***; ***Harris et al., 2021***). Together, these results suggest that the Trabid patient muta-tions produce hypomorphic variants impaired in two distinct biochemical activities: polyubiquitin hydrolysis and STRIPAK-binding.

### Decreased cell density in the brains of mice harboring Trabid patient mutations

Given the microcephaly observed in children with Trabid mutations, and to understand the conse-quence of Trabid hypomorphic variants in vivo, we examined knock-in mice carrying Trabid patient mutations for evidence of brain development abnormalities (***Figure 2A***). Mice heterozygous or homo-zygous for the R438W or A451V mutation were viable, fertile, and born at expected Mendelian ratios (***Figure 2B***). Because germline Trabid knockout or C443S homozygous knock-in mice exhibit peri-natal lethality (***Dickinson et al., 2016***) (our unpublished data), these new R438W and A451V mutant mouse strains allow us to interrogate Trabid function in vivo. Homozygous mutant mice from the R438W colony weighed on average ~15% less than wild-type littermates, while R438W heterozygous and A451V mutant mice did not exhibit significant weight loss (***Figure 2C***). Immunohistochemical analysis of brain sections from weaned littermate mice of both mutant strains revealed the normal structure and laminar organization of the cerebral cortex, but intriguingly a reduction in cell number

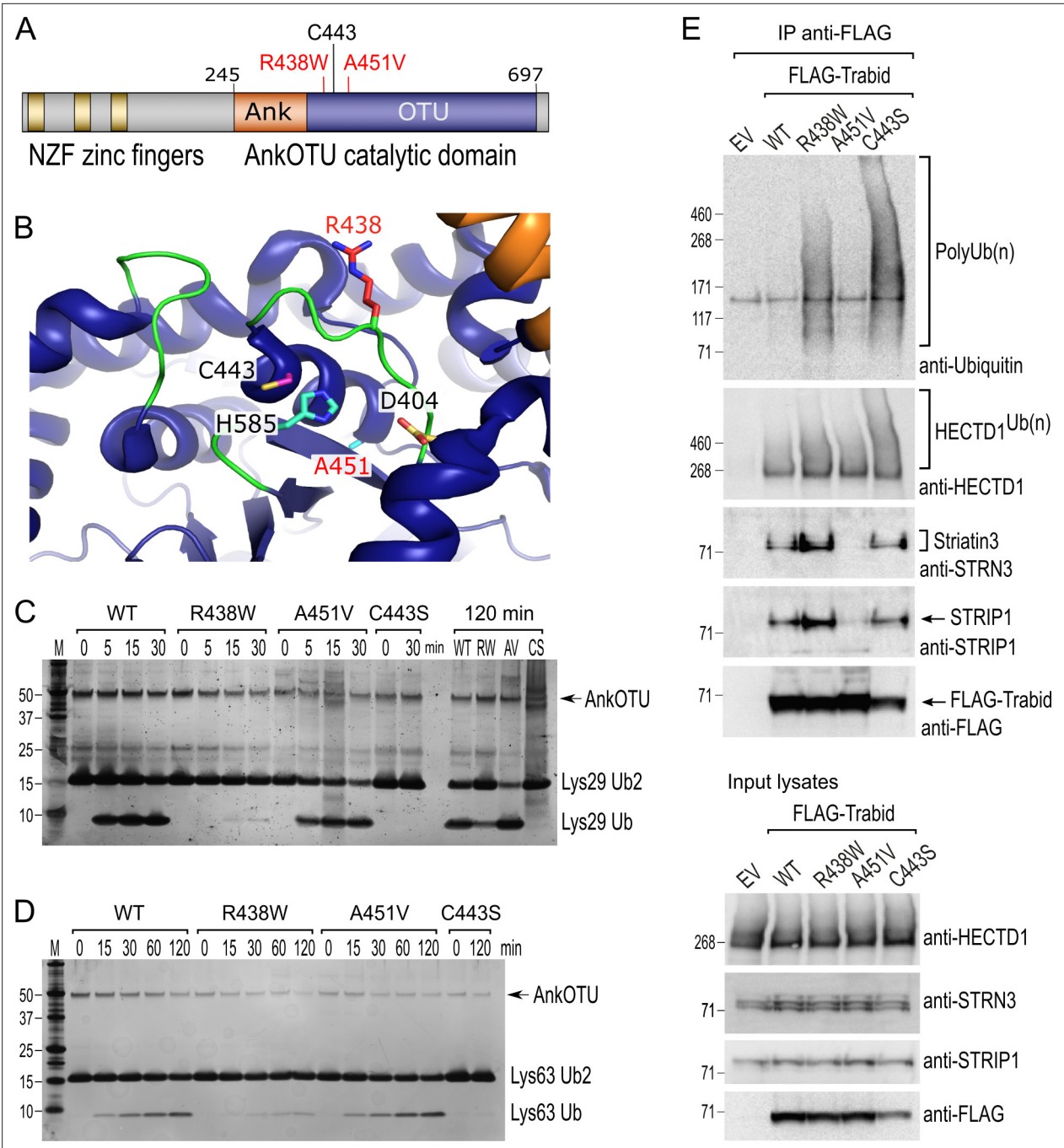

**Figure 1.** Trabid patient mutants are impaired in deubiquitylating (DUB) activity and Striatin-interacting phosphatase and kinase (STRIPAK) binding. (**A**) Trabid patient mutations R438W and A451V flank on either side of the catalytic cysteine C443 in Trabid's AnkOTU domain. (**B**) Residue R438 projects prominently into the catalytic cleft formed by the Ank and ovarian tumor (OTU) domain, whereas residue A451 resides at the back of the active site. The positions of the catalytic triad residues C443, H585, and D404 are shown. Crystal structure of AnkOTU domain: PDB 3ZRH. (**C** and **D**) In vitro DUB assays. Hydrolysis of Lys29-linked (**C**) and Lys63-linked (**D**) di-ubiquitin chains by purified wild-type and mutant Trabid AnkOTU proteins. The Trabid R438W AnkOTU domain exhibits strongly reduced DUB activity. M, molecular weight markers in kilodaltons. (**E**) Immunoprecipitation of FLAG-Trabid from the lysates of transfected HEK293T cells showed that FLAG-Trabid R438W and C443S mutants co-precipitated high amounts of endogenous polyubiquitin (PolyUb(n) smear), indicating impaired DUB activity. FLAG-Trabid A451V failed to efficiently co-precipitate Striatin3 and STRIP1, indicating loss of binding to the STRIPAK complex. All FLAG-Trabid proteins co-precipitated similar amounts of the E3 ubiquitin ligase HECTD1. EV, empty vector control.

The online version of this article includes the following source data and figure supplement(s) for figure 1:

**Source data 1.** Uncropped images of the silver-stained gels depicted in *Figure 1C and D*.

*Figure 1 continued on next page*

*Figure 1 continued*

**Source data 2.** Uncropped images of the western blots depicted in *Figure 1E*.

**Figure supplement 1.** Thermal stability and catalytic cysteine reactivity of Trabid mutant AnkOTU domains.

**Figure supplement 1—source data 1.** Uncropped images of the silver-stained gel depicted in *Figure 1—figure supplement 1B*.

**Figure supplement 2.** Replicate of the Lys29-linked di-ubiquitin deubiquitylating (DUB) assay using the same purified AnkOTU protein preparations used in *Figure 1C*.

**Figure supplement 2—source data 1.** Uncropped images of the silver-stained gel depicted in *Figure 1—figure supplement 2*.

was apparent in different brain regions of R438W and A451V homozygous mice compared to wild-type littermate controls (*Figure 2D*). The number of Ctip2+ medium spiny neurons in the striatum of R438W and A451V mutant mice was reduced compared with wild-type littermates, and homozygous mice showed a greater cell number loss than heterozygous mice, suggesting a gene dosage effect (*Figure 2E*). Moreover, R438W and A451V mutant mice had a reduced number of Olig2+ oligodendrocytes in the forebrain and midbrain compared to wild-type littermate mice (*Figure 2F*). Olig2+ cell numbers in homozygous brain sections were consistently strongly reduced (>30%) compared to wild-type mice, independent of age or sex. Of a combined 16 sets of littermate mice from both R438W and A451V colonies, reduced neuronal or glial cell numbers were conspicuous in homozygous brain sections of 12 littermate sets (*Figure 2E and F*; see also *Figure 3*), indicating incomplete penetrance of the mutant phenotype. Collectively, these results suggest that Trabid's DUB and STRIPAK-binding activities are required to produce the correct numbers of neuronal and glial cells in the developing brain.

## Trabid mutant mice exhibit a motor deficit consistent with reduced number of dopaminergic neurons and projections

Given the cell number deficits in the midbrain of Trabid mutant mice and the reported midbrain neurodegeneration and locomotor defects in Trabid mutant *Drosophila* (*Kounatidis et al., 2017*), we asked if the number of dopaminergic neurons that control motor function might be affected in Trabid R438W and A451V mutant mice. Midbrain and striatal coronal sections were immunostained for Tyrosine Hydroxylase (TH), the rate-limiting enzyme for dopamine synthesis and a marker of dopaminergic neurons. A reduction in the number of TH+ neurons was observed in the substantia nigra pars compacta (SNc) of Trabid mutant mice from both R438W and A451V colonies, independent of age or sex (*Figure 3A*). SNc neurons project axons to the striatum to produce an extensive network of axonal terminals that communicate with striatal neurons (*Matsuda et al., 2009*). Consistent with the lack of prominent TH+ neuronal processes emanating from the SNc neurons that remain in the mutant midbrain, TH immunoreactivity was reduced in the striatum of homozygous R438W and A451V mutant mice (*Figure 3B and C*). Also in mutant brain sections, the reduced abundance and intensity of TH+ neuronal processes were readily apparent in cortical regions including the motor cortex and claustrum that regulate motor responses (*Figure 3B and C*). Differences in TH staining in the mutant striatum compared to the wild-type were modest or not observed in all littermates examined (*Figure 3D*), indicating incomplete penetrance of the mutant Trabid alleles.

Given the altered abundance of TH+ cells and projections in motor-related brain regions of Trabid mutant mice, we evaluated the motor function of these mice using a rotarod assay. Homozygous R438W mice showed a significantly reduced latency to fall at 20, 25, 30, and 35 RPM, compared to controls (*Figure 3E*), as did homozygous A451V animals at 30 and 35 RPM (*Figure 3E*). This indicates that Trabid patient mutations impair motor coordination, consistent with a deficiency in dopaminergic neurons. The smaller size of R438W homozygous mice compared to wild-type littermates (*Figure 2C*) could be a contributing factor to the compromised rotarod performance of these mutants. However, given that A451V homozygous mice on average are not smaller than wild-type littermates, the motor deficit exhibited by both R438W and A451V homozygous mice is likely caused by the abnormal cell numbers and reduced dopaminergic neurons in their brains (*Figures 2 and 3*). Taken together, these results suggest that Trabid regulates the brain cellular architecture and neuronal circuitry required for normal motor function.

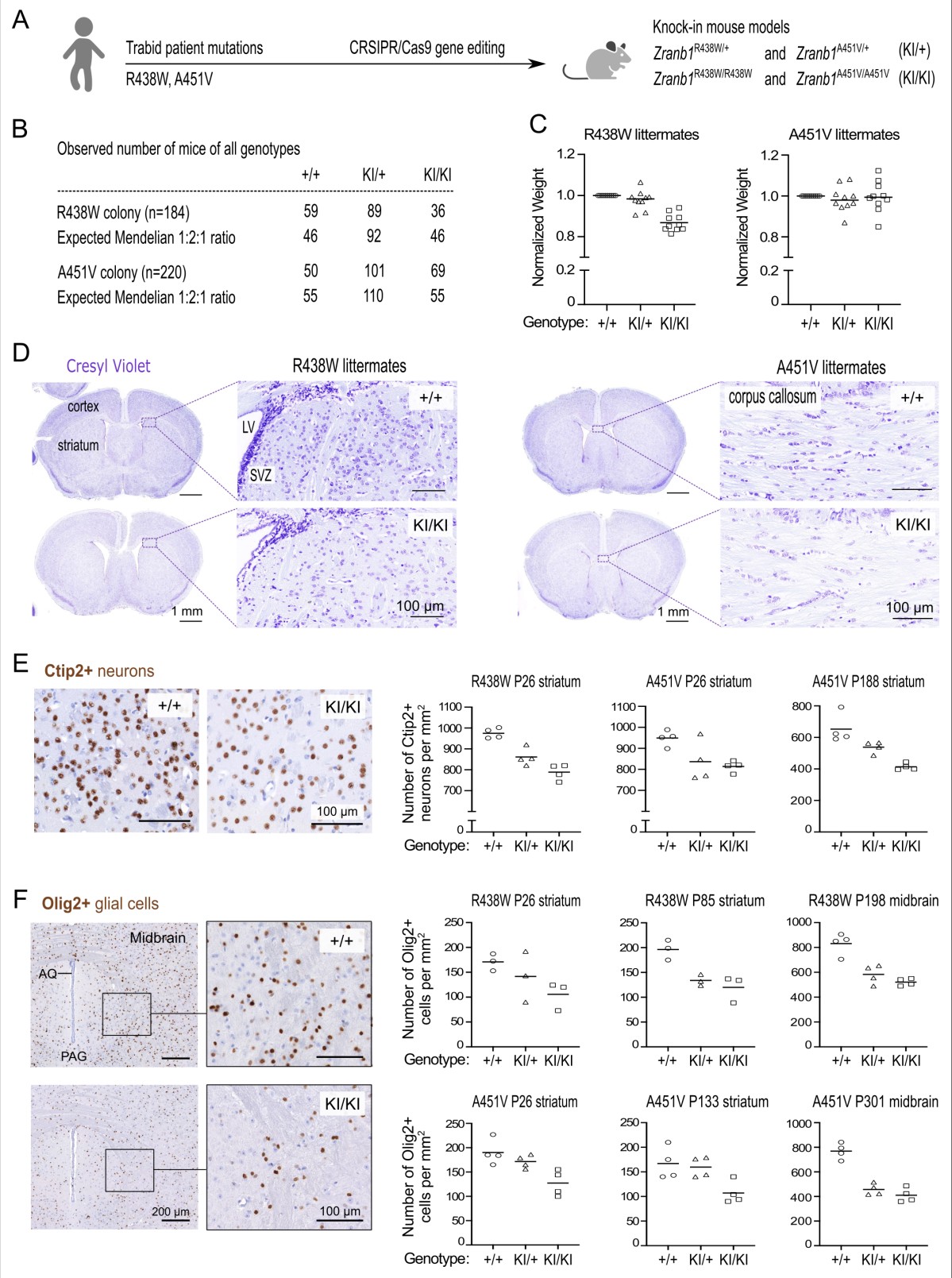

**Figure 2.** Decreased cell density in the brains of mice harboring Trabid patient mutations. (**A**) Schematic of knock-in mouse strains harboring *Zranb1/Trabid* patient mutations generated by CRISPR/Cas9 genome editing. (**B**) Numbers of mice of all genotypes produced from heterozygous intercrosses in the R438W or A451V colony, respectively. (**C**) Normalized weight of 10 sets of weaned littermate mice (five males, five females; age range P26-P222) from each Trabid mutant colony. For each set of littermates, the weight of the wild-type (+/+) mouse was set at 1, and the weight of heterozygous (KI/+)

*Figure 2 continued on next page*

*Figure 2 continued*

or homozygous (KI/KI) mutant mice was expressed as a ratio of the wild-type mouse weight. (**D**) Coronal brain sections (Bregma ± 0.3 mm) of weaned (postnatal day P26) littermate males from the R438W and A451V mouse colonies were stained with Cresyl Violet to assess general anatomy and cellular organization in the cerebral cortex and striatum. LV, lateral ventricle; SVZ, subventricular zone. An overall decrease in Cresyl Violet staining intensity was evident in mutant homozygous (KI/KI) sections from both mouse strains compared to similar sections from the respective wild-type (+/+) littermate. (**E**) Ctip2 IHC revealed reduced numbers of medium spiny neurons in the striatum of mutant mice from the R438W and A451V colonies compared to wild-type littermate mice. Representative images shown are coronal brain sections of P26 male littermates from the R438W colony. Each data point represents the cell count in a randomly selected, non-overlapping 1 mm square area in the striatum from both brain hemispheres. The age and brain region of the cell count for three sets of littermate mice from the indicated mutant strain are specified. Cell numbers were quantified blinded to genotype using images processed in Fiji software. See *Figure 2—figure supplement 1*. (**F**) Olig2 IHC revealed reduced numbers of oligodendrocytes in the striatum and midbrain of homozygous mice from the R438W and A451V colonies compared to respective wild-type littermates. Representative images shown are coronal sections from the ventral midbrain of P301 female littermates from the A451V colony (AQ, cerebral aqueduct; PAQ, periaqueductal gray). Each data point represents the cell count in a randomly selected, non-overlapping 1 mm square area in the striatum or midbrain from both brain hemispheres. The age and brain region of the cell count for three sets of littermate mice from the indicated mutant strain are specified. Cell numbers were quantified blinded to genotype using images processed in Fiji. See *Figure 2—figure supplement 1*.

The online version of this article includes the following figure supplement(s) for figure 2:

**Figure supplement 1.** Workflow for quantification of cell numbers in Immunohistochemistry (IHC) images.

## Trabid patient mutants fail to efficiently limit APC ubiquitylation in cells

To understand the molecular basis for the neuronal and behavioral phenotypes of Trabid mutant mice, we focused on APC— an established regulator of mammalian brain development (*Yokota et al., 2009*; *Preitner et al., 2014*) and a protein that we have shown becomes strongly ubiquitylated upon loss of Trabid or Striatin (*Tran et al., 2008*; *Tran et al., 2013*). Moreover, given that Striatin binds directly to APC (*Breitman et al., 2008*), we hypothesized that the Trabid patient mutants impaired in DUB activity or Striatin-binding would impact APC ubiquitylation and function in cells. To test this, we generated cells with doxycycline (dox)-inducible expression of FLAG-Trabid and examined the ubiquitylation status of endogenous APC in dox-treated cells. The levels of ubiquitin-modified APC were strongly suppressed in cells expressing wild-type FLAG-Trabid (*Figure 4*). In contrast, induction of the FLAG-Trabid R438W mutant, like the catalytically dead FLAG-Trabid C443S, did not repress APC ubiquitylation, and induction of FLAG-Trabid A451V expression only partially reduced APC ubiquitylation compared to control cells (*Figure 4*). These results suggest that both DUB and STRIPAK-binding activities of Trabid are required for efficient deubiquitylation of APC in cells.

## Trabid patient mutants impede EGFP-APC transport to the leading edge of migrating cells

APC decorates the membrane protrusions at the leading edge of migrating cells (*Näthke et al., 1996*) and we have shown that Trabid is a key regulator of APC localization to these cortical structures (*Tran et al., 2013*). Consistent with this insight, FLAG-Trabid localized prominently with EGFP-APC in the tips of long cell processes (*Figure 5A*). Analysis of several of these tip clusters by super-resolution microscopy revealed ~30% overlap between the FLAG-Trabid and EGFP-APC signals, indicating that a significant fraction of these proteins co-localize in cell tips (*Figure 5B*). In sub-confluent cultures, EGFP-APC localized strongly to the lamellipodial leading edge of the majority of cells co-transfected with wild-type FLAG-Trabid (*Figure 5C and D*). By contrast, in cells co-transfected with the FLAG-Trabid mutants R438W and A451V, EGFP-APC formed irregular punctae at sub-cortical regions close to the leading edge, and EGFP-APC aggregates are often sequestered to punctae formed by the R438W FLAG-Trabid mutant (*Figure 5C and D*). Such sites may contain ubiquitylated substrates bound to DUB-defective Trabid mutants, as we have previously demonstrated with Trabid C443S (*Tran et al., 2008*; *Licchesi et al., 2011*). Consistently, FLAG-Trabid C443S co-localized with EGFP-APC in punctae near the tips of long cell processes (*Figure 5E*). The Trabid substrate HECTD1 was sequestered to C443S punctae in the cytosol, but intriguingly not to punctae residing in proximity of the leading edge membrane (*Figure 5E*). In near-confluent cultures following prolonged co-transfection of FLAG-Trabid mutants and EGFP-APC, we observed the striking phenomena of strong EGFP-APC aggregation on or near the plasma membrane and thin tube-like processes extending between neighboring cells decorated with abundant EGFP-APC aggregates of various sizes (*Figure 5F*). In cells co-transfected with wild-type FLAG-Trabid, EGFP-APC is concentrated in cell tips (*Figure 5F*). Collectively, these results

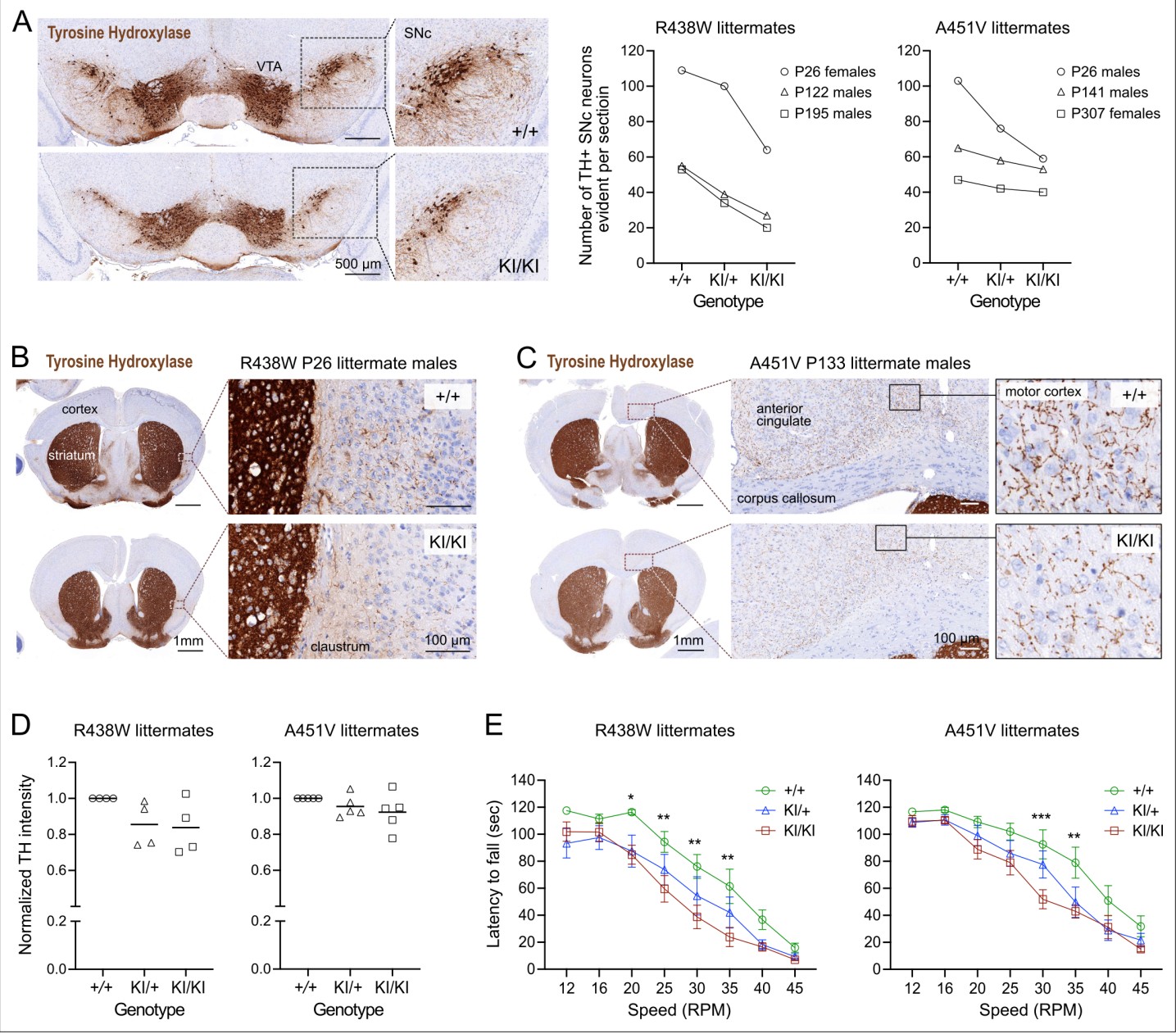

**Figure 3.** Trabid mutant mice exhibit a motor deficit consistent with reduced number of dopaminergic neurons and projections. (**A**) Tyrosine Hydroxylase (TH) Immunohistochemistry (IHC) of coronal midbrain sections revealed reduced numbers of dopaminergic neurons in the substantia nigra pars compacta (SNc) of homozygous mutant mice from the Trabid R438W and A451V colonies compared to similar midbrain sections of the respective wild-type littermates. Representative images shown are midbrain sections of P26 female littermates from the R438W colony. VTA, ventral tegmental area. Numbers of TH+ SNc neurons were quantified using IHC images processed in Fiji. See *Figure 2—figure supplement 1*. (**B**) Tyrosine Hydroxylase IHC of coronal brain sections (Bregma ± 0.3 mm) of P26 littermate mice from the Trabid R438W colony revealed reduced TH staining intensity in the striatum and fewer TH+ projections in the adjacent cortex of homozygous mutant mice compared to similar regions in the wild-type littermate (magnified area). (**C**) Tyrosine Hydroxylase IHC of coronal brain sections (Bregma ± 0.3 mm) of P133 littermate mice from the Trabid A451V colony revealed reduced abundance and intensity of TH+ projections in several cortical regions including the primary/secondary motor cortex of homozygous mutant mice compared to similar regions in the wild-type littermate (magnified area). (**D**) Tyrosine Hydroxylase staining intensity in the striatum of mutant mice from the R438W and A451V colonies normalized to the staining intensity of wild-type littermate sections. IHC signals were measured as the optical density of the region of interest demarcated manually using Fiji. Each symbol represents one mouse of the indicated genotype belonging to a set of littermate mice from the R438W colony (n=4 sets: P26 males, P93 females, P307 males, P480 females) or the A451V colony (n=5 sets: P26 males, P41 females, P133 males, P188 females, P303 females). (**E**) Rotarod performance of 3–4 month-old littermate mice from the R438W and A451V colonies. Each data point represents the average latencies of eight mice (four males, four females), where each mouse was subjected to six trials over 3 days. Wild-type (+/+), heterozygous (KI/+), and homozygous (KI/KI) littermate mice were tested together. The experimenter was blinded to genotype. A repeated measures

*Figure 3 continued on next page*

*Figure 3 continued*

two-way ANOVA and Dunnett's multiple comparisons test were applied to the data, using wild-type mice as the control group. Asterisks denote statistically significant differences between wild-type (+/+) and homozygous (KI/KI) mutant mice. *$p<0.05$, **$p<0.01$, ***$p<0.001$. Error bars, ± SEM.

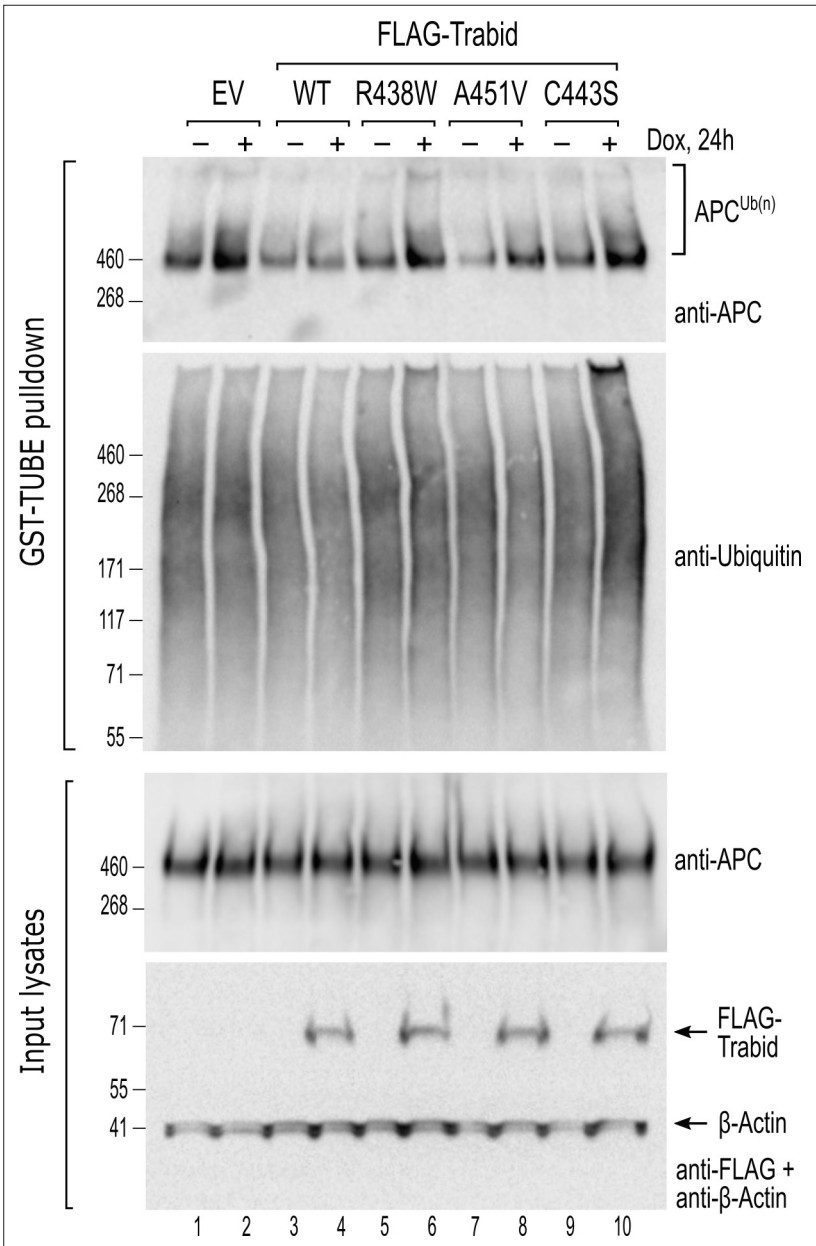

**Figure 4.** Trabid patient mutants fail to efficiently suppress adenomatous polyposis coli (APC) ubiquitylation in cells. HEK293 cells stably transfected with doxycycline-inducible constructs for the expression of wild-type or mutant FLAG-Trabid were untreated (-) or treated with dox (+; 100 ng/mL) for 24 h. Cell lysates were subjected to a GST-TUBE pulldown to enrich the ubiquitylated proteome. Precipitated material and input lysates were processed for Western blotting using the indicated antibodies. Endogenous APC protein levels remained unchanged irrespective of FLAG-Trabid expression (input lysates), but ubiquitylated APC species were only efficiently suppressed by wild-type FLAG-Trabid expression (APC$^{Ub(n)}$; WT +). EV, empty vector control.

The online version of this article includes the following source data for figure 4:

**Source data 1.** Uncropped images of western blots depicted in *Figure 4*.

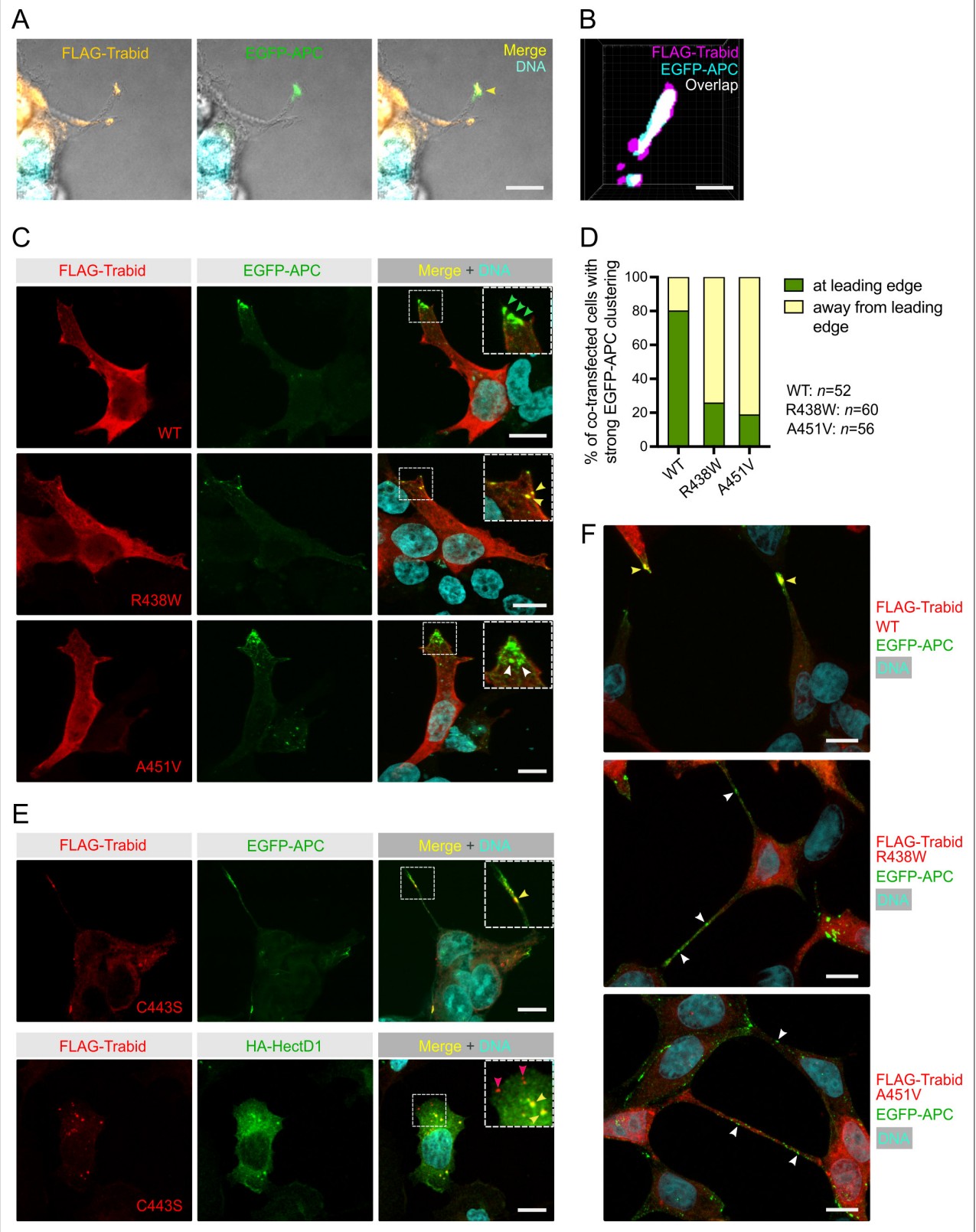

**Figure 5.** Trabid patient mutants impede the transport of EGFP-APC to the leading edge of migrating cells. (**A**) Combined differential interference contrast and immunofluorescence images of HEK293T cells extending long cell processes that contain FLAG-Trabid and EGFP-APC in the cortical protrusions of co-transfected cells (arrowhead in merged image). Scale bar, 10 μm. (**B**) 3D-SIM super-resolution microscopy analysis of HEK293T cell tip clusters (n=5) revealed ~30% overlap of FLAG-Trabid and EGFP-APC signals at resolutions of ~100 nm in *xy* and 320 nm in *z*. Scale bar, 100 nm. (**C**)

*Figure 5 continued on next page*

*Figure 5 continued*
EGFP-APC membrane clusters at the leading edge of migrating HEK293T cells (green arrowheads) are readily detected with FLAG-Trabid wild-type (WT) co-expression. In cells expressing FLAG-Trabid patient mutants, EGFP-APC aggregate in punctate structures near the leading edge (white arrowheads), that are often sequestered together with the FLAG-Trabid R438W mutant (yellow arrowheads). Scale bar, 10 µm. (**D**) Quantification of strong EGFP-APC clustering at or away from the leading edge membrane of co-transfected HEK293T cells, as shown in (**C**). Numbers of cells analyzed from three independent transfections are shown. (**E**) In co-transfected HEK293T cells, DUB-dead FLAG-Trabid C443S form puncta that sequesters EGFP-APC near the tips of long cell protrusions or sequesters HA-HECTD1 in the cytoplasm (yellow arrowheads). Notably, HA-HECTD1 was not detected in FLAG-Trabid C443S punctae that form near the leading edge membrane of migrating cells (red arrowheads). Scale bar, 10 µm. (**F**) In near-confluent HEK293T cells 48 h post-transfection, FLAG-Trabid patient mutants induce long thin tubes that connect neigboring cells and extensive EGFP-APC aggregations are conspicuous along these tubular structures. This phenomenon was not observed in cells transfected with wild-type FLAG-Trabid. Scale bar, 10 µm.

indicate that Trabid's DUB and STRIPAK-binding activities are required for the efficient localization of APC to plasma membrane sites involved in polarised cell migration.

## Trabid patient mutations perturb the axonal trafficking of APC-tdTomato and impair the trajectory of neurite outgrowth

APC regulates the dynamic interactions of the microtubule and actin cytoskeletons at cortical membranes to direct the formation of neuronal growth cones and the trajectory of axon growth (*Dogterom and Koenderink, 2019*). To investigate the effect of patient Trabid mutations on APC localization in primary neurons, we generated mice expressing an APC-tdTomato fluorescent protein under the control of the endogenous *Apc* gene promoter (*Figure 6A*). APC-tdTomato mice were bred with knock-in mice carrying either the Trabid R438W or A451V mutant allele (*Figure 2*) and neural progenitors were isolated at embryonic day E11.5 from compound heterozygotes. We then performed live cell imaging to analyze the trafficking of APC-tdTomato in neurites extending from progenitors undergoing differentiation (*Figure 6A*). Endogenous Trabid protein levels were comparable in neural progenitor cultures derived from embryos bearing wild-type or R438W and A451V mutant Trabid alleles (*Figure 6B*). In neurons with wild-type Trabid, APC-tdTomato accumulated strongly in the tips of growing neurites, marking prominent growth cones that drive axon elongation (*Figure 6C*; *Zranb1*$^{+/+}$). In neurons with heterozygous Trabid R438W or A451V mutation, APC-tdTomato exhibited broad distribution along the length of the neurite, and the formation of growth cones was severely compromised (*Figure 6C*, *Zranb1*$^{R438W/+}$). Tracking analysis of the turn angles of APC-tdTomato intensities revealed that neurites with wild-type Trabid elongated in a polarized manner, whereas neurites with R438W or A451V mutant Trabid turned back or retracted at a greater frequency (*Figure 6D and E*). These data suggest that Trabid's DUB and STRIPAK-binding activities are required for the efficient transport of APC to neurite tips to promote the formation of growth cones required for polarized axon elongation.

## Discussion

We describe a novel function for the deubiquitylating enzyme Trabid as a key regulator of axonal growth and guidance that likely underpins the neurodevelopmental defects observed in children with Trabid mutation. Our data suggests that Trabid's mechanism of action is to suppress the ubiquitylation of APC to regulate its intracellular trafficking. A hypoubiquitylated APC pool is efficiently localized to the cortical cytoskeleton where it directs neuronal growth cone formation and polarized axon elongation. The identification of human Trabid variants that disrupt the distribution of APC to the leading edge of migrating cells provides a plausible explanation for the associated patient neurodevelopmental disorders. Our work identifies Trabid's DUB- and STRIPAK-binding activities, and the control of APC localization, as crucial events during embryonic and neural development.

The two patients carrying Trabid missense mutations R438W and A451V were diagnosed with a range of distinct congenital disorders including craniofacial abnormalities, seizures, developmental delay, autism—and both patients presented with microcephaly and constipation (*Deciphering Developmental Disorders, 2015*). We propose that the underlying cause of these seemingly unrelated conditions, broadly classified as neurocristopathies, is the abnormal specification or migration of neural crest cells in the developing embryo (*Vega-Lopez et al., 2018*). Discrete neural crest populations contribute to the development of craniofacial structures, the forebrain and midbrain, and the enteric

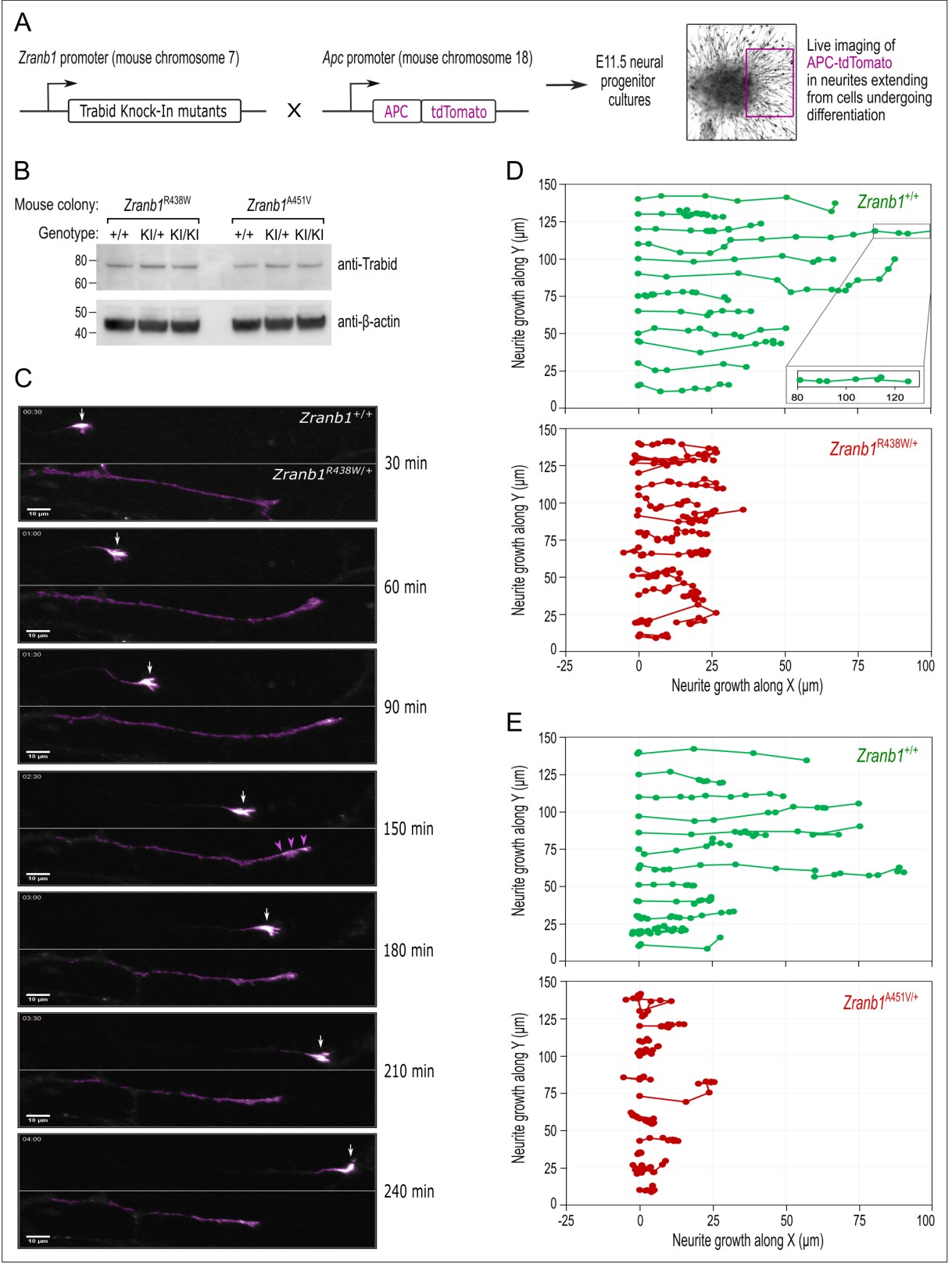

**Figure 6.** Trabid patient mutations perturb the trafficking of APC-tdTomato and the trajectory of neurite outgrowth. (**A**) Schematic of outcross between knock-in mice harboring *Zranb1/Trabid* patient mutations and *Apc*^tdTomato^ mice, where the APC-tdTomato fluorescent protein is expressed under the control of the endogenous *Apc* gene promoter. Midbrain neural progenitors derived from E11.5 embryos of these outcrosses were used for live cell imaging of APC-tdTomato trafficking in growing neurites. (**B**) Western blot analysis of Trabid protein expression in E11.5 midbrain neural progenitors

*Figure 6 continued on next page*

*Figure 6 continued*

derived from heterozygous intercrosses in the *Zranb1* R438W or A451V mouse colonies. Mutant Trabid protein levels are expressed at comparable levels to wild-type Trabid protein (compare KI/KI to +/+). (**C**) Live cell imaging of APC-tdTomato in neurites extending from E11.5 midbrain neural progenitors of a *Zranb1^R438W/+^*; *Apc^tdTom/+^* outcross. The frames shown are from a 4 h time-lapse portraying the movement of APC-tdTomato. APC-tdTomato accumulated in the tips of neurites with wild-type Trabid (*Zranb1^+/+^*) to generate growth cones that drive axon elongation (white arrows). In neurons with mutant Trabid (*Zranb1^R438W/+^*; *Zranb1^A451V/+^* data not shown), APC-tdTomato intensities were broadly distributed along neurites (purple arrowheads) and the formation of growth cones was abrogated. Scale bar, 10 μm. (**D** and **E**) The movement of Apc-tdTomato in individual neurite tips was tracked to visualize the growth trajectory. Neurites extending from E11.5 midbrain neural progenitors of *Zranb1^R438W/+^*;*Apc^tdTom/+^* (**D**) or *Zranb1^A451V/+^*; *Apc^tdTom/+^* (**E**) outcrosses were analyzed. APC-tdTomato fluorescence synchronized with the movement of a membrane dye during neurite extension and retraction (data not shown) and was, therefore, used as a proxy for neurite growth along an arbitrary X and Y plane. Each data point represents the location of APC-tdTomato at 15 min intervals of a 4 h time-lapse. Neurite tracking data were compiled from the imaging of neurite outgrowth in three independent neurosphere cultures established from E11.5 embryos of three independent *Zranb1^R438W/+^*; *Apc^tdTom/+^*, or *Zranb1^A451V/+^*; *Apc^tdTom/+^* outcrosses. All image acquisition, processing, and analyses were performed blinded to genotype.

The online version of this article includes the following source data for figure 6:

**Source data 1.** Uncropped images of western blots depicted in *Figure 6B*.

nervous system (**Anderson et al., 2006**; **Creuzet et al., 2006**). Therefore, errors in the formation or polarized migration of neural crest cells could account for all the patient phenotypes, including the reduction in brain volume (microcephaly). Presently, we can only speculate that the irregular cellular architecture and reduced cell numbers in the brains of mice harboring Trabid patient mutations (**Figures 2 and 3**) reflect the errant migration of neural crest or neural progenitor cell populations in early embryogenesis (**Silva et al., 2019**). While Trabid mutant mice did not exhibit microcephaly, they showed a motor deficit (**Figure 3**) consistent with the locomotor defects reported for Trabid loss-of-function in *Drosophila* (**Kounatidis et al., 2017**). Intriguingly, the fly Striatin homolog CKA is involved in axonal transport and motor coordination (**Neisch et al., 2017**), and *Drosophila* Strip1 regulates endosomal trafficking and axon elongation (**Sakuma et al., 2014**). Endosomal trafficking defects have been linked to reduced proliferation of neural progenitors and microcephaly (**Carpentieri et al., 2022**). Three independent groups have identified Trabid's association with STRIPAK (**Sowa et al., 2009**; **Tran et al., 2013**; **Harris et al., 2021**). We have now established the functional significance of this interaction. The requirement of Trabid binding to STRIPAK for efficient protein trafficking, polarized axon growth, and motor coordination (**Figures 3, 5 and 6**), supports the view that Trabid-STRIPAK regulates an evolutionarily conserved mechanism of cell movement required for normal brain development and establishment of the correct neuronal circuitry in the motor system.

The impairment of two distinct Trabid functions—polyubiquitin hydrolysis and STRIPAK-binding—led to common cellular, developmental, and behavioral phenotypes (**Figures 2–6**; **Deciphering Developmental Disorders, 2015**), strongly suggesting that these activities act in the same pathway. We propose that APC is the primary molecular target of Trabid action in cytoskeleton organization and polarized axon growth (**Figure 7**). APC is an established regulator of cell adhesion and migration, and it governs the cortical actin and microtubule cytoskeleton dynamics required to form and steer axonal growth cones (**Dogterom and Koenderink, 2019**; **Efimova et al., 2020**). In a yeast two-hybrid screen, we found that Trabid and Striatin interacted with the armadillo repeat domain (ARD) of APC, but not with an APC ARD mutant that caused cell-cell adhesion defects (**Hamada and Bienz, 2002**; **Tran et al., 2008**). This implies that a functional interaction between Trabid, Striatin, and APC promotes the fidelity of cell-cell or cell-substratum contacts. Our model of Trabid action in **Figure 7** integrates published data showing that (1) Trabid complexes with Striatin/STRIPAK in human cell lines (**Sowa et al., 2009**; **Tran et al., 2013**; **Harris et al., 2021**), and (2) Striatin binds directly to the ARD domain of APC (**Breitman et al., 2008**). We propose that STRIPAK recruits Trabid to deubiquitylate APC. This allows APC to accumulate at the leading edge of migrating cells to promote efficient, polarized locomotion (**Figure 7A**). The Trabid R438W mutant still binds STRIPAK and is recruited to APC, but it is impaired in its ability to cleave ubiquitin chains from APC. The Trabid A451V mutant retains full DUB activity but it cannot be recruited to APC via STRIPAK. Both mutant scenarios cause APC to become persistently modified with ubiquitin chains that lead to APC delocalization from cell tips and defective cell movement (**Figure 7B**). Consistent with this model, Trabid and/or Striatin deficiency caused APC hyperubiquitylation and aggregation, perturbed actin assembly and microtubule stability,

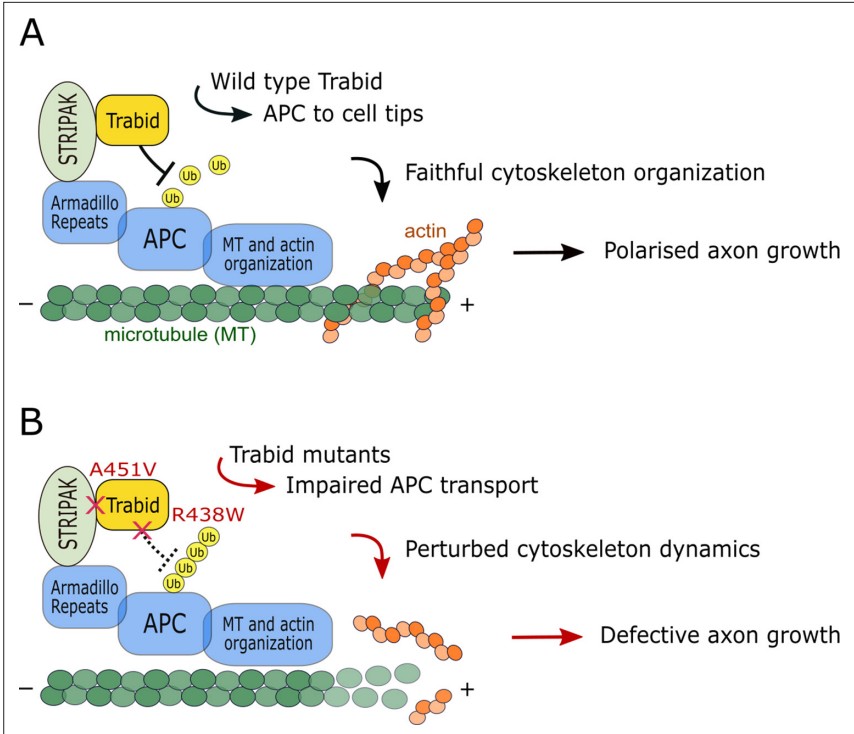

**Figure 7.** Model of Trabid's mechanism of action in axon growth and guidance. (**A**) Striatin-interacting phosphatase and kinase (STRIPAK) recruits Trabid to deubiquitylate adenomatous polyposis coli (APC) (and possibly other substrates). This promotes efficient APC accumulation at microtubule plus ends to coordinate the actin and microtubule cytoskeleton dynamics that drive directional cell migration and polarized axon growth. (**B**) In cells expressing Trabid mutants that are DUB-impaired (R438W) or STRIPAK-binding-deficient (A451V), APC becomes persistently modified with ubiquitin chains that retard its transport to cell tips. This impairs cytoskeleton organization which leads to defective axon elongation and cell migration. Future studies will aim to identify the mechanism of Trabid recruitment to STRIPAK, the ubiquitin-modified residue(s) on APC (and possibly other substrates), and the composition of polyubiquitin linkages on substrates including the Lys29- and Lys33-linked atypical chain types preferentially targeted by Trabid.

and inhibited the migration of mouse and human cell lines (*Bai et al., 2011*; *Tran et al., 2013*) (our unpublished data).

We have shown that ubiquitin-modified APC correlated with its binding to Axin in the β-catenin destruction complex (*Tran and Polakis, 2012*), whereas non-ubiquitin-modified APC accumulates in membrane protrusions (*Tran et al., 2013*). Thus, reversible modification with polyubiquitin could be the long-hypothesized molecular switch that regulates the distribution of APC between its many functional pools in cells (*Dikovskaya et al., 2001*; *Bienz, 2002*). Chronic APC ubiquitylation in Trabid deficient/mutant cells might result in increased APC sequestration into Axin destruction complexes or promote spurious interactions with ubiquitin-binding proteins that retard APC trafficking along microtubules. Rescue of the APC transport defect could hypothetically be achieved by inhibition of glycogen synthase kinase 3 (GSK3), which we have shown abolishes APC ubiquitylation (*Tran and Polakis, 2012*). Supporting this idea is the finding that GSK3 inactivation promotes the association of APC with microtubule plus-ends to drive polarized cell migration (*Etienne-Manneville and Hall, 2003*).

Optimal cell migration and adhesion require stable, acetylated microtubules (*Aguilar et al., 2014*; *Bance et al., 2019*). Of note, the loss of APC and the STRIPAK proteins STRIP1 and CTTNBP2 strongly reduced the acetylated microtubule network in neurons (*Yokota et al., 2009*; *Shih et al., 2014*; *Sakuma et al., 2015*). Furthermore, Striatin depletion perturbed cell-cell adhesion (*Breitman et al., 2008*; *Lahav-Ariel et al., 2019*) and axon elongation defects caused by STRIP1 mutation are linked to the dysregulation of neuronal adhesion (*Sakuma et al., 2014*). It would be interesting to investigate if chronic APC ubiquitylation and dysfunction underlie all Trabid and STRIPAK mutant phenotypes.

## Materials and methods

### Protein purification and characterization

Wild-type (WT), R438W, A451V, and C443S Trabid AnkOTU domains (245-697) were expressed in BL21 competent *E. coli* and purified as described previously (*Licchesi et al., 2011*) with minor modifications. *E. coli* cells were induced with 300 mM IPTG and grown overnight at 16°C. Cells were lysed by sonication in 50 mL lysis buffer (20 mM Tris pH 8.5, 200 mM NaCl, 10% glycerol, 1 mM PMSF, 2 x Roche protease inhibitor tablets, 3 mM $MgCl_2$, 0.1 mg/mL DNAse, 150 µL lysozyme, 10 mM β-mercaptoethanol) per 2 L culture. Anion-exchange chromatography (MonoQ 5/50) was performed using the ÄKTA pure system at 4°C. See *Figure 1—figure supplement 1* for the characterization of the thermal stability and ubiquitin reactivity of purified Trabid AnkOTU proteins. The thermal stability of purified WT, R438W, A451V, and C443S AnkOTU recombinant proteins was assessed using Tycho NT.6 (NanoTemper) following the manufacturer's protocol. The ubiquitin suicide probe assay using ubiquitin-propargylamine (Ub-PA) was performed as previously described (*Gersch et al., 2017*). For each reaction, 0.5 µM of purified WT or mutant Trabid AnkOTU protein was mixed with 5 µM of Ub-PA and 5 mM DTT. The reaction was incubated at 37°C for 1 h and stopped by the addition of SDS sample buffer (50 mM Tris-HCl pH 8, 10% v/v glycerol, 2% w/v SDS, 0.01% w/v bromophenol blue, 2.5% v/v 2-mercaptoethanol). Assays were resolved by SDS-PAGE and visualized by silver staining (Silver Stain Plus; Biorad). For DUB assays, a total of 20 µL reaction consisting of 0.25 µM purified AnkOTU domain was incubated with 1 µM of Lys29- or Lys63-linked di-ubiquitin chains (R&D Systems) in DUB reaction buffer (50 mM NaCl, 50 mM Tris pH 7.4, and 5 mM DTT) at 37°C. Reactions were stopped at the indicated times by the addition of SDS sample buffer and visualized by SDS-PAGE and silver staining.

### Cells, plasmids, and antibodies

An inducible lentiviral CRISPR–Cas9 system (*Aubrey et al., 2015*) was used to overexpress Trabid in *Figure 4*. PCR amplified DNA encoding human wild-type or mutant Trabid with an N-terminal FLAG tag was cloned into the EcoRI/NheI sites of a doxycycline-inducible pFTREtight MCS rtTAadvanced puro lentiviral vector (*Brumatti et al., 2013*). Transduced HEK293 cells were selected for puromycin resistance and pooled clones were used for experiments. Transient transfection of plasmids pcDNA3.1-FLAG-Trabid (*Tran et al., 2013*), FLAG-Trabid mutants generated by site-directed mutagenesis (Quik-Change, Agilent), pEGFP-C1-APC (*Rosin-Arbesfeld et al., 2001*), and pCMV-HA-HECTD1 (*Sarkar and Zohn, 2012*) in HEK293T cells was performed using Lipofectamine 2000 (Invitrogen). HEK293T cells (ATCC) and HEK293 cells (Cellbank Australia) were authenticated by STR profiling and frequently tested for mycoplasma contamination. Cells were cultured in DMEM supplemented with 10% FBS and 2 mM L-Glutamine in a humidified incubator at 37 °C with 5% $CO_2$. Mouse monoclonal Trabid antibodies and rabbit polyclonal HectD1 antibodies have been described (*Tran et al., 2013*). Commercial antibodies were purchased from a variety of vendors: anti-APC ALi 12–28 (Santa Cruz, sc-53165), anti-Striatin (BD Biosciences, 610838), anti-Striatin3 (Novus Biologicals, NB110-74572), anti-Strip1 (Abcepta, AP2817b), anti-ubiquitin P4D1 (Cell Signaling, mAb#3936), anti-FLAG M2 (Sigma, F3165), anti-β-actin-HRP (Sigma, A3854), anti-β-tubulin (Sigma, T8328), anti-HA (Roche, 3F10), anti-Tyrosine Hydroxylase (Millipore, AB152), anti-Ctip2 (Abcam, ab18465), and anti-Olig2 (Abcam, ab109186). Western blot signals were developed using enhanced chemiluminescence reagents (Clarity, Bio-Rad).

### Protein enrichment from cell lysates

Confluent cells in 10 cm dishes were lysed on ice in 1 mL lysis buffer containing 1% IGEPAL CA-630, 50 mM Tris-HCl, pH 7.5, 120 mM NaCl, 1 mM EDTA, and protease inhibitor tablets (Roche). Lysates were centrifuged at 14,000 rpm for 10 min to remove cell debris. The supernatant was assayed for protein (BCA kit, Pierce), and then 1 mg total protein was immunoprecipitated with anti-FLAG M2 antibody and protein G sepharose in a total volume of 1 mL lysis buffer. Enrichment of polyubiquitin chains was performed using purified GST-TUBE (Tandem Ubiquitin Binding Entity) protein (*Hjerpe et al., 2009*). A volume of cell supernatant containing 1 mg total protein was diluted 1:1 with 4 M urea and then incubated with 10 µg GST-TUBE and Glutathione sepharose 4B. Immunoprecipitations and GST-TUBE pulldowns were performed overnight with constant rotation at 4 °C. Protein complexes were washed twice with lysis buffer and once with PBS, then eluted at 95 °C for 5 min in Laemmli buffer for Western blotting analysis.

## Mice

All mouse studies were conducted according to the Australian Code for the care and use of animals for scientific purposes and complied with relevant ethical regulations approved by the Walter and Eliza Hall Institute Animal Ethics Committee (ref# 2018.050 and 2020.028). The $Zranb1^{R438W}$, $Zranb1^{A451V}$, and $Apc^{tdTomato}$ knock-in mice were generated on a C57BL/6J background using CRISPR–Cas9-mediated gene editing by the Melbourne Advanced Genome Editing Centre (MAGEC) at the Walter and Eliza Hall Institute. To generate a R438W mutation within the $Zranb1$ gene on mouse chromosome 7, a single guide (sg) RNA of the sequence GAC TAT ATG CAC TTT GGA AC was used to create double-stranded breaks within the $Zranb1$ locus to stimulate homologous recombination and an oligo donor of the sequence TAT AAA CTG GTC TTT GGA GTT GGC TAC ACG TCT GGA CAG TAG ACT ATA TGC ACT TTG GAA CTG GAC TGC CGG AGA TTG TTT ACT TGA CTC AGT ACT ACA AGC TAC ATG GGG CAT TTA TGA CAA A was used to introduce the R438W mutation. The sgRNA and donor sequence along with Cas9 mRNA were injected into the cytoplasm of fertilized one-cell stage embryos generated from wild-type C57BL/6J breeders. To generate the A451V mutation, an sgRNA of the sequence ACT CAG TAC TAC AAG CTA CA and an oligo donor of the sequence ACA GTA GAC TAT ATG CAC TTT GGA ACC GGA CTG CCG GAG ATT GTT TAC TTG ACT CAG TAC TAC AAG TCA CAT GGG GCA TTT ATG ACA AAG ACT CGG TGC TTC GGA AAG CCC TGC ATG ACA GCC TG CAT was used. Twenty-four hours later, two-cell stage embryos were transferred into the uteri of pseudo-pregnant female mice. Viable offspring were genotyped by next-generation sequencing. Targeted animals were backcrossed twice to wild-type C57BL/6J to eliminate off-target mutations. Generation of C57BL/6J mice expressing the APC-tdTomato fusion protein under the control of the endogenous $Apc$ promoter (B6J.$Apc^{tdTom}$) was based on methods previously described (**Ng et al., 2020**). Briefly, a sgRNA of the sequence AGA CGT CAC GAG GTA AGA CC was used to create double-stranded breaks within the $Apc$ locus to stimulate homologous recombination. A targeting vector containing homology arms of ~1.4 kilobases was used to introduce the tdTomato coding sequence after the last $Apc$ coding exon. Forward (ACC TGT TCC TGT ACG GCA TG) and reverse (GCC TCC CAA AAT GAC CAG TG) primers to detect the tdTomato sequence were used to screen viable pups for integration of the targeting vector by PCR.

## Neurite outgrowth from cultured neural progenitors

To generate mouse embryos expressing Apc-tdTomato with wild-type or mutant Trabid, mice heterozygous for the Apc-tdTomato allele ($Apc^{tdTom/+}$) were outcrossed with mice heterozygous for the Trabid R438W or A451V mutant allele ($Zranb1^{R438W/+}$ or $Zranb1^{A451V/+}$). Embryos at E11.5 were harvested from pregnant females and ventral midbrains were dissected as previously described (**Thompson and Parish, 2013**). The isolated midbrain tissue was enzymatically dissociated in Hank's Balanced Salt Solution containing 0.05% trypsin and 0.1% DNase I for 12 min at 37 °C. Cells in the tissue were separated by mechanical dissociation, counted, and then plated in serum-free, N-2 medium consisting of a 1:1 mixture of Ham's F12 and Minimum Essential Medium supplemented with 15 mM HEPES buffer, 1 mM glutamine, 6 mg/mL glucose, 1 mg/mL bovine serum albumin, and N-2 supplement (Gibco). Cells were seeded at a density of 250,000 cells per well in a 24-well plate at 37 °C, 3% $CO_2$. Following 48 h, an aliquot of the resultant neurosphere cultures were sequenced to ascertain genotypes. To differentiate midbrain progenitors towards the dopaminergic lineage, neurospheres were resuspended in N-2 medium containing BDNF and GDNF (Gibco; 30 ng/mL each) then seeded in glass bottom ibidi chamber slides pre-coated with poly-D-lysine and laminin (Sigma; 10 µg/mL each) for live imaging of neurite outgrowth as described below.

## Histology and IHC analysis

Mice were euthanized by $CO_2$ inhalation and intracardial perfusion was performed to fix the tissues. Perfusion was initiated with Dulbecco's phosphate-buffered saline followed by 10% neutral buffered formalin (NBF). After perfusion, the brain and tissues were dissected and post-fixed in 10% NBF for 24 h before paraffin-embedding with the Tissue-Tek VIP 6 AI automated tissue processor (Sakura Finetek). Formalin-fixed paraffin-embedded tissues were sectioned into 7 µm slices using a microtome and mounted on positively charged slides. Immunohistochemistry (IHC) was performed with the Omnis Auto-immunostaining platform using Agilent EnVision Target retrieval solution and the optimal dilution of the primary antibody against the target protein. Secondary antibody detection

was performed with the Dako EnVision+ Single Reagents HRP and FLEX DAB+ Substrate Chromogen System. Slides were counterstained with hematoxylin. Sections from wild-type, heterozygous, and homozygous littermate mice were mounted and stained on the same slide to ensure IHC signals could be directly compared. Cell counts and IHC signal intensities were quantified using Fiji software (NIH). The workflow for counting cells from IHC images is described in *Figure 2—figure supplement 1*. IHC signal intensities were measured as the optical density proportional to the concentration of the stain. Briefly, the color spectra of the DAB- and the hematoxylin-stained image was separated by color deconvolution, converted to grayscale, thresholding was applied, and the mean pixel intensity of the region of interest was measured.

### Rotarod performance test

The motor coordination of 3- to 4-month-old littermate mice from Trabid R438W and A451V colonies was measured using a rotating rod (Rotamex-5, Columbus Instruments). Mice were lowered onto a 3 cm diameter rod rotating at 12, 16, 20, 25, 30, 35, 40, or 45 revolutions per min (RPM) for 2 min or until they fell. Animals were given a 5 min rest between RPM sessions. Two trials were performed with a 1 h break between trials. The latency to fall was recorded for each RPM. A 1 s penalty was added if an animal failed to walk in time with the rod but rather gripped the rod and rotated with it (cartwheel). Animals underwent two trials per day across 3 days.

### Microscopy and image analysis

*Confocal microscopy* – Cells grown in Lab-Tek II chamber slides were fixed with 4% paraformaldehyde for 10 min at room temperature, then permeabilized with 0.2% Triton-X100 and blocked with 5% normal goat serum (NGS). Cells were incubated with primary antibodies diluted in 5% NGS overnight at 4 °C, followed by Alexa Fluor-conjugated secondary antibodies for 1 h at room temperature. ProLong Gold antifade reagent with DAPI (Invitrogen) was used to mount coverslips to microscope slides. Confocal images were acquired on a Zeiss LSM 880 Airyscan microscope using a 63 x/1.4 N.A. oil objective, and 405, 488, and 594 nm lasers. Maximum intensity projections of raw images comprising 10–15 *z* sections were created using Fiji.

*3D structured illumination microscopy* – Super-resolution three-dimensional structured illumination microscopy (3D-SIM) was performed on the DeltaVision OMX-SR system using a 60 x/1.42 N.A. PlanApo oil immersion objective, sCMOS cameras, and 488, 568, and 640 nm lasers. 3D-SIM images consisted of 15 raw images per focal plane per color channel with 125 nm between each z-step. Images were reconstructed and the color channels were aligned using the reconstruction and alignment algorithms in softWoRx 7.0. Fraction of overlap between FLAG-Trabid and EGFP-APC was measured in Imaris 9.2.

*Lattice lightsheet time-lapse imaging and analysis* – Live cell time-lapse imaging of APC-tdTomato fluorescence in growing neurites was acquired using the Zeiss Lattice Light Sheet 7 microscope. A 561 nm laser formed a light sheet of length 30 μm with a thickness of 1 μm at the sample plane via a 13.3 x, 0.44 NA objective. tdTomato fluorescence was collected via a 44.83 x, 1 NA detection objective lens. Data was collected with a frame rate of 60 ms and a z-step of 300 nm. Each region was imaged at 15 min intervals for 4–6 h. Fluorescence was collected via a multi-band stop, LBF 405/488/561/633, filter. Images were subsequently deskewed using Zeiss's Zen 3.4 software. Samples were measured at 37 °C and 5% $CO_2$. Maximum intensity projections were created from the deskewed data in Fiji. Neurite tips were tracked using Trackmate (v6.0.3) and LoG detector with 12 μm diameter, 0.08 threshold, no filters, and subpixel localization turned on. A minimum of five frames were set for track inclusion. Turn angles were calculated using Matlab R2019b. Three points were used to calculate the turn. The first two points determine the direction of neurite growth. The angle between this direction to the third point determines the turn angle of the neurite. The turn angle was normalized such that 0° constitutes a step straight forward with no turn and 180° is a turn backwards.

## Acknowledgements

We are indebted to Prof David L Vaux for his generous support and mentorship in the early stages of this study. We thank Drs S Wilcox, S Scutts, S Cobbold, J Heath, D Newgreen, J Murphy, J Vince, J Silke, C Parish, K Scicluna, M Herold, V Wimmer, T Thomas, LE Ohman, W Wong, K Newton, and V Dixit for discussion, reagents and support; and WEHI's Bioservices staff for outstanding animal care.

Part of the funding for this study was provided by the Australian National Health and Medical Research Council (Ideas Grant #1181580 to EV, HT, and BMT). GD was supported by a fellowship from the Bodhi Education Fund. Work in the authors' laboratory is made possible by operational infrastructure grants through the Australian Government Independent Research Institutes Infrastructure Support (IRISS) and the Victorian State Government OIS. The generation of all mutant and transgenic mice used in this study was supported by Phenomics Australia and the Australian Government through the National Collaborative Research Infrastructure Strategy (NCRIS) program.

## Additional information

### Competing interests

David Komander: Founder, shareholder and serves on the SAB of Entact Bio. The other authors declare that no competing interests exist.

### Funding

| Funder | Grant reference number | Author |
| --- | --- | --- |
| National Health and Medical Research Council | 1181580 | Elizabeth Vincan<br>Hoanh Tran<br>Bang Tran |
| Bodhi Education Fund | Fellowship | Grant Dewson |

The funders had no role in study design, data collection and interpretation, or the decision to submit the work for publication.

### Author contributions

Daniel Frank, Maria Bergamasco, Michael J Mlodzianoski, Data curation, Formal analysis, Investigation, Writing – review and editing; Andrew Kueh, Ellen Tsui, Data curation, Investigation, Methodology; Cathrine Hall, Data curation, Investigation; Georgios Kastrappis, Data curation, Formal analysis; Anne Kathrin Voss, Catriona McLean, Maree Faux, Kelly L Rogers, Formal analysis; Bang Tran, Formal analysis, Funding acquisition; Elizabeth Vincan, Formal analysis, Supervision, Funding acquisition; David Komander, Resources, Formal analysis, Writing – review and editing; Grant Dewson, Resources, Formal analysis, Supervision, Funding acquisition, Visualization, Writing – review and editing; Hoanh Tran, Conceptualization, Resources, Data curation, Formal analysis, Supervision, Funding acquisition, Investigation, Visualization, Writing – original draft, Project administration

### Author ORCIDs

Daniel Frank (iD) http://orcid.org/0000-0003-4998-2220
Maria Bergamasco (iD) http://orcid.org/0000-0003-3322-9701
Michael J Mlodzianoski (iD) https://orcid.org/0000-0002-3510-9167
Ellen Tsui (iD) https://orcid.org/0009-0005-7340-7412
Cathrine Hall (iD) http://orcid.org/0009-0005-1004-9435
Anne Kathrin Voss (iD) http://orcid.org/0000-0002-3853-9381
Catriona McLean (iD) https://orcid.org/0000-0002-0302-5727
Maree Faux (iD) https://orcid.org/0000-0001-7770-6683
Kelly L Rogers (iD) http://orcid.org/0000-0002-6755-0221
Bang Tran (iD) https://orcid.org/0000-0002-3108-8805
Elizabeth Vincan (iD) https://orcid.org/0000-0002-8607-4849
Hoanh Tran (iD) https://orcid.org/0000-0002-0176-4112

### Ethics

All mouse studies were conducted according to the Australian Code for the care and use of animals for scientific purposes and complied with relevant ethical regulations approved by the Walter and Eliza Hall Institute Animal Ethics Committee (ref# 2018.050 and 2020.028).

Reviewer #1 (Public Review): https://doi.org/10.7554/eLife.90796.3.sa1

Reviewer #2 (Public Review): https://doi.org/10.7554/eLife.90796.3.sa2
Author Response https://doi.org/10.7554/eLife.90796.3.sa3

## Additional files

### Supplementary files
• MDAR checklist

### Data availability
All data generated or analysed during this study are included in the manuscript and supporting files.

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
