## [Editor Report · eLife assessment]

This study defines the roles for two different missense mutations observed in patients in the Trabid/ZRANB1 gene associated in children with a range of congenital disorders including reduced brain size. The study is **important** because the findings have theoretical or practical implications beyond a single subfield, as the study of DUB and cytoskeletal alterations have implications for neurodevelopment broadly. The methods are **convincing** as they utilize appropriate and validated methodology in line with current state-of-the-art by incorporating knock-in mice of the patient mutations. Many of the reviewer comments were focused on potential next experiments, rather than on evaluation of the data at hand, and the authors have considered these as future studies. The work as presented suggests critical roles for Trabid in the STRIPAK complex mediating APC deubiquitylation.

---

## [Referee Report · Reviewer #1 (Public Review)]

In this work, Frank, Bergamasco, Mlodzianoski et al study two microcephaly-associated patient variants in TRABID to identify and characterize a previously unrecognized role of this deubiquitylation enzyme during neurodevelopment. The authors generate TRABID p.R438W and p.A451V knock in mice, which exhibit smaller neuronal and glial cell densities as well as motor deficits, phenotypes that are consistent with the congenital defects observed in the patients. Through in vitro and cellular immunoprecipitation assays, the authors demonstrate that the p.R438W variant impairs the K29- and K63-chain cleavage activity of TRABID, while the p.A451V variant reduces binding to the STRIPAK complex, a previously identified TRABID interactor with established functions in cytoskeletal organization and neural development. Ubiquitylation assays performed in HEK293T cells further reveal that the hypomorphic patient variants are deficient in deubiquitylating APC, a previously identified substrate of TRABID that has been shown to control the neuronal cortical cytoskeleton during neurite outgrowth. Ex vivo experiments provide evidence that axonal APC trafficking and neurite outgrowth is disturbed in differentiating neural progenitors isolated from mouse embryos carrying Trabid patient alleles. From these experiments the authors propose a model in which TRABID- and STRIPAK-dependent APC deubiquitylation regulates its axonal trafficking to ensure faithful neurite outgrowth and misregulation of this function leads to neurodevelopmental phenotypes in TRABID/ZRANB1 patients.

---

## [Referee Report · Reviewer #2 (Public Review)]

Although Trabid missense mutations are identified across a range of neurodevelopmental disorders, its role in neurodevelopment is not understood. Here the authors study two different patient mutations and implicate defects in its deubiquitylating activity and interactions with STRIPAK. Knockin mice for these mutations impaired trafficking of APC to microtubule plus ends, with consequent defects in neuronal growth cone and neurite outgrowth.

The authors focus on R438W and A451V, two missense mutations seen in patients. Recombinant fragments showed R438W is nearly completely DUB-dead whereas A451V showed normal activity but failed to efficiently precipitate STRIPAK. Knockin of these mutations showed a partially penetrant reduced cortical neuronal and glial cell numbers and reduced TH+ neurons and their neuronal processes. Cell culture demonstrated that both DUB and STRIPAK-binding activities of Trabid are required for efficient deubiquitylation of APC in cells, and alter APC transport along neurites. APC-tdTomato fluorescent reporter mice crossed with the Trabid mutants confirmed these results. The results suggest that Trabid's mechanism of action is to suppress APC ubiquitylation to regulate its intracellular trafficking and neurite formation.

---

## [Author Response]

The following is the authors’ response to the original reviews.

We thank the reviewers for their time and effort to review our manuscript. We have provided a response to their thoughtful questions below. In our revised manuscript, we have expanded the Discussion to comment on the significance of reversible modification of APC with polyubiquitin and how the APC transport defect might be rescued. We have also included Figure 1—figure supplement 2 and Figure 1—source data 1, to address the minor concerns of Reviewer #1.

**Reviewer #1 (Recommendations For The Authors):**
To address the weaknesses outlined below, I have the following comments and suggestions for experiments:1. Functional link between mouse phenotypes and proposed mechanism: could the authors rescue neuron/glia cell density or motor defects by restoring axonal trafficking of APC?

We have shown that inhibition of glycogen synthase kinase 3 (GSK3) abolished APC ubiquitylation (PMID 22761442). Etienne-Manneville and Hall have reported that GSK3 inactivation promotes APC association with microtubule plus ends to drive polarised astrocyte migration (PMID 12610628). It is therefore conceivable that treating Trabid mutant neurons with a GSK3 inhibitor could suppress APC ubiquitylation, restore APC transport, and rescue the defective axon growth. GSK3 has multiple targets so there are caveats to using potent inhibitors of this kinase. But such an experiment is integral to a future study aimed at rescuing Trabid mutant mouse phenotypes by GSK3 inhibition.

Does perturbation of APC trafficking phenocopy the defects of TRABID p.R438W and p.A451V knock in mice during neurodevelopment? I appreciate that these experiments might not be easily feasible.

Presently we do not know how to directly perturb APC transport (besides generating a Trabid mutation). Speculatively, APC phosphosite mutants which mimic constitutive phosphorylation by GSK3 might accumulate polyubiquitin, aggregate, and exhibit disrupted axonal transport. We predict that such APC mutants will cause neurodevelopmental abnormalities in mouse models.

Thus, alternatively, could the authors provide evidence from unbiased proteomic approaches that APC is a major substrate of TRABID- and STRIPAK-dependent deubiquitylation during neurodevelopment? E.g., what are the changes in the ubiquitylome of neural progenitor cells isolated from mouse embryos with TRABID mutant alleles and is APC amongst the top dysregulated hits? What are the changes in the interactome of TRABID p.A451V and is the STRIPAK complex a major interactor that is lost?

We are generating antibodies capable of immunoprecipitating endogenous Trabid from mouse cells. This antibody tool will allow us to characterise the Trabid-STRIPAK complex using advanced ubiquitin proteomic approaches to determine interactors and changes to the ubiquitylome of Trabid mutant cells.

2. Related to the point 1, given that TRABID has been reported to be a regulator of immune signaling pathways (PMID: 26808229, 37237031), can the authors exclude a contribution of this function to the observed phenotypes during neurodevelopment?

We have not observed any cellular or tissue phenotypes in young or aged Trabid mutant mice indicative of immune system dysregulation. We and others have shown that Trabid deficiency has no impact on the transcription of interferon and NF-κB-stimulated genes or cytokine production in mouse and human cells (PMID 18281465; 17991829; unpublished). Nevertheless, a formal investigation is required to determine any changes to immune signalling pathways in our Trabid mutant mice.

3. Based on previously published interactions, the authors propose that TRABID uses the STRIPAK complex to recruit its substrate APC. Could the authors provide experimental evidence for this by using their cellular model in Figure 4? Would depleting components of the STRIPAK complex in HEK 293T cells stably transfected with DOX-inducible WT-TRABID stabilize APC ubiquitylation upon dox induction?

We have demonstrated that RNAi-mediated depletion of all 3 striatin proteins in HEK293T cells increased the levels of ubiquitin-modified APC (PMID 23277359). Moreover, depleting Trabid and the 3 Striatins together strongly increased the ubiquitin-modified APC pool, consistent with our model that Trabid and STRIPAK function together to deubiquitylate APC. In our inducible system, we would likely need to *eliminate* the expression of the STRIPAK component that directly recruits Trabid to achieve a null effect of Trabid overexpression on APC deubiquitylation. Experiments are in progress to determine which STRIPAK component binds directly to Trabid.

4. Related to point 3, given that A451, the residue that mediates STRIPAK binding is in close proximity to the catalytic cysteine residue, how do the authors envision STRIPAK binding and OTU-dependent cleavage activity to work together at a structural level?

A451 resides at the back of the active site in a pocket hypothesised to accommodate a short peptide from an interacting protein. The A451V mutant AnkOTU domain purified from bacteria retained full DUB activity, suggesting that Trabid’s ability to cleave polyubiquitin is independent of its ability to bind STRIPAK. Striatin proteins contain WD40 repeats which is a protein fold that binds ubiquitin (PMID 21070969). While the DUB- and STRIPAK-binding activities of Trabid might not be coupled structurally, it is plausible that Striatin could modulate Trabid’s ubiquitin linkage specificity in cells through allosteric interactions with the ubiquitin chain on the substrate.

5. Is it known why APC needs to be reversibly modified with ubiquitin to be transported in axons and how increased APC ubiquitylation leads to impaired transport or could the authors speculate on this?

We have shown that APC ubiquitin modification correlated with its binding to Axin in the β-catenin destruction complex (PMID 22761442). Conversely, non-ubiquitin-modified APC accumulates in membrane protrusions (PMID 23277359). From this we have proposed that ubiquitin regulates the distribution of APC between its two major functional pools in cells. Chronic APC ubiquitylation in Trabid deficient/mutant neurons might result in increased APC sequestration into Axin destruction complexes and/or promote spurious interactions with ubiquitin binding proteins that cause APC to aggregate, and therefore retard its transport in axons.

Additional minor comments to consider:• Figure 1C: What are the protein smears in the in vitro assays of A541V 15min and CS 120min? I would assume that contaminants from the protein preparations should be the same across different conditions and in particular across different time points of the same Trabid mutant.

In replicate DUB assays using the same AnkOTU protein preparations we did not detect any smears (Figure 1—figure supplement 2). It is unclear what caused the smears in Figure 1C, but it is plausible that contaminants in specific tubes/assays are contributing factors.

• Figure 1D: why is the amount of AnkOTU protein reduced for WT, R438W, and A541 in a time-dependent manner?

With increasing incubation time in DUB assays, adducts of various molecular weights may form between ubiquitin and the AnkOTU domain. It is plausible that some of these adducts are non-gel-resolved high molecular weight aggregates that sequester some of the AnkOTU proteins. These aggregates, which could have been retained in the loading wells, were presumably washed away during our silver staining procedure hence we do not see them in the full-length gel (Figure 1—source data 1).

**Reviewer #2 (Recommendations For The Authors):**
The partial penetrance of the mouse knockin phenotype is confusing, especially as this is evident on an apparently inbred background. Can authors explain the factors that contribute to these differences?

Low mutant Trabid protein expression in distinct neural crest or progenitor populations could contribute to the reduced penetrance of the cell number phenotype. APC dysfunction in Trabid mutant cells might also impact its role as a negative regulator of the Wnt signalling pathway which regulates neuronal and glial cell fates in the developing brain (PMID 9845073). It is conceivable that in some Trabid mutant mice where APC dysfunction is mild (due to low levels of mutant Trabid protein expression), compensatory mechanisms overcome APC’s reduced functions in Wnt signalling and cytoskeleton organization to permit normal brain development. A future study to investigate perturbations of Wnt signalling pathways in Trabid mutant mice is warranted.

The use of the term 'hemizygous' is confusing, as it typically refers to when one copy of a gene is present as in X-linked conditions. Might the authors mean 'heterozygous'?

All instances of ‘hemizygous’ in the manuscript have been amended to ‘heterozygous’.

Fig. 3A y-axis units is confusing. Do the authors mean number of TH+ SNc neurons evident per section?

We have amended the y-axis in Fig. 3A to indicate number of TH+ neurons evident per section.

Since the TH phenotype is one of the phenotypes that is partially penetrant, did authors include both penetrant and non-penetrant mice in Fig. 3 and other figures? Shouldn't there be error bars in Fig. 3A, since multiple mice were presumably used for analysis for each condition?

Each data point in Figure 3A represents one mouse in a set of littermate mice with the indicated age, sex, and genotype. Generating midbrain SNc sections at similar bregma positions across wild-type and mutant littermate brains for accurate IHC comparison proved challenging. Unanticipated technical issues limited the quantification of equivalent midbrain sections to three sets of littermate mice from each respective R438W or A451V mutant colony. The cell number reduction is more obvious in some mutants than others, but the effect is observed across all ages and gender, providing confidence that the phenotype is robust. In Figure 2 we have included only mutant mice with clearly fewer brain cells than wild-type littermates. We have not performed comprehensive IHC analysis of brains from all the mice used for the rotarod assay in Figure 3E, but predict that mutant mice have a spectrum of neural/glial cell deficits in one or more brain areas that adversely impacted the motor circuitry causing their impaired motor function.